

# Assessment of degenerative changes in the atlanto-odontoid joint using cone-beam computed tomography (CBCT) imaging

Ramazan Berkay Peker[1], Omer Said Sezgin[2], Saadettin Kayipmaz[2] and Senem Tuğra Dönmez[2]

[1] Department of Dentomaxillofacial Radiology, Faculty of Dentistry, Trakya University, Edirne, Turkey
[2] Department of Dentomaxillofacial Radiology, Faculty of Dentistry, Karadeniz Technical University, Trabzon, Turkey

## ABSTRACT

**Background.** This study sought to evaluate the severity of osteoarthritis in the atlanto-odontoid joint (AOJ) using cone-beam computed tomography (CBCT) and analyse its distribution by age and sex.

**Materials and Methods.** The CBCT images of 215 patients (130 females, 85 males; aged 18–80 years) taken for dental purposes were retrospectively analysed. The severity of the osteoarthritis in the AOJ was graded from 0 to 3, while its relationships with age and sex were assessed via ordinal and binary logistic regression analyses.

**Results.** Osteoarthritis-related changes were detected in 64.7% of patients (62.3% of females, 68.3% of males). Females showed 44.3% decreased odds of severe osteoarthritis compared to males ($p = 0.041$). Individuals aged 18–29 showed 99.9% decreased odds of severe osteoarthritis, those aged 30–39 showed 99.7% decreased odds, those aged 40–49 showed 99.4% decreased odds, those aged 50–59 showed 97.1% decreased odds, and those aged 60–69 showed 96.0% decreased odds of severe osteoarthritis compared to individuals aged 70–80 (all $p < 0.01$). Each additional year of age increased the risk of osteoarthritis by 8.0% (OR: 1.080; $p < 0.001$) (odds ratio (OR): 1.080; $p < 0.001$), although it was not significantly associated with sex ($p = 0.248$).

**Conclusion.** CBCT is effective in detecting degenerative changes in the AOJ. Moreover, it is associated with lower radiation doses than conventional CT. Ageing significantly increases both the probability and the severity of osteoarthritis. Routine AOJ evaluation during CBCT imaging may significantly enhance clinical decision-making by enabling the early detection and management of osteoarthritis.

# INTRODUCTION

The cervical spine plays a critical role in supporting the head, facilitating multidirectional motion, and protecting the upper spinal cord. Among its segments, the upper cervical spine provides substantial mobility—particularly rotational motion—while simultaneously

Corresponding author
Ramazan Berkay Peker,
rberkaypeker@trakya.edu.tr

maintaining stability through specialized osseoligamentous structures. This rotational capacity is primarily enabled by the atlantoaxial joint, which connects the first cervical (C1) and second cervical (C2) vertebrae (*Smoker, 1994*; *Swartz, Floyd & Cendoma, 2005*). The median part of this joint, which is known as the atlanto-odontoid joint (AOJ), is formed by the odontoid process of the C2 vertebra, the anterior arch of the C1 vertebra and the transverse ligament complex (*Standring, 2016*; *Liu et al., 2014*). The AOJ and lateral parts of the atlantoaxial joint provide 40–70% of the total cervical spine rotation (*Betsch et al., 2015*; *Lakshmanan et al., 2005*). This extensive range of motion, concentrated primarily at the C1–C2 level, predisposes the AOJ to chronic mechanical loading and wear, thereby increasing its susceptibility to osteoarthritic degeneration (*Mayer et al., 2023*).

Degenerative changes in the AOJ appear radiographically similar to those found in other synovial joints and commonly present with osteophytes, subchondral cysts, cortical sclerosis and narrowing of the joint space (*Liu et al., 2014*; *Genez et al., 1990*; *Scutellari et al., 2007*; *Zapletal et al., 1997*). The AOJ is a critical anatomical structure responsible for head and neck rotation, and osteoarthritic changes in this joint are associated with clinical outcomes such as headache, restricted neck movement and an increased predisposition to odontoid process fractures (*Zapletal et al., 1997*; *Brolin, 2003*; *Olerud et al., 1999*). Osteophyte formation in the atlantoaxial joint can lead to compression of adjacent neurovascular structures, particularly the vertebral artery, resulting in symptoms such as cervicogenic headaches. Recent studies have demonstrated that surgical decompression of the vertebral artery alleviates these symptoms, highlighting the clinical significance of this anatomical relationship (*Sonfack et al., 2024*). Degenerative changes in the AOJ are increasingly observed with advancing age. *Liu et al. (2014)* reported that the prevalence of atlantoaxial osteoarthritis reached 93% in individuals over 70 years of age, supporting the view that this condition represents a growing public health concern in aging populations.

In recent years, cone-beam computed tomography (CBCT) has become increasingly prevalent in dental and maxillofacial imaging due to its three-dimensional (3D) imaging capability and lower radiation exposure compared to conventional CT. Despite this, most radiographic assessments of AOJ osteoarthritis in the literature have primarily relied on conventional CT or magnetic resonance imaging (MRI) (*Liu et al., 2014*; *Zapletal et al., 1997*; *Zapletal et al., 1995*).

While gadolinium-enhanced MRI is widely acknowledged for its diagnostic precision, it is often impractical for routine screening owing to its high cost, limited accessibility, and invasive nature. By contrast, CBCT provides a more accessible and less invasive alternative, with sufficient resolution to detect early osseous changes. Moreover, morphometric methods have frequently been utilized in the literature to classify joint space narrowing in degenerative joint diseases, further supporting the utility of imaging-based assessments in this context (*Nayak, Bhatnagar & Pai Khot, 2024*).

Compared to conventional CT and MRI, CBCT offers a significantly reduced radiation dose, cost-efficient operation, and is widely available in dental settings, making it a practical tool for incidental and routine assessments (*Jaju & Jaju, 2015*).

In addition to its established utility in evaluating temporomandibular joint degeneration, CBCT has also proven effective in detecting degenerative changes in the adjacent cervical

spine. Evidence supports its diagnostic value in early-stage degenerative joint disease and its potential application in the craniocervical region, including the AOJ (*Nayak, Bhatnagar & Pai Khot, 2024*). Furthermore, gray values derived from the second and third cervical vertebrae have been shown to significantly correlate with bone mineral density in postmenopausal women, suggesting a broader diagnostic potential of CBCT in assessing systemic skeletal health (*Slaidina et al., 2022*).

To our knowledge, no previous study has specifically evaluated the severity of AOJ osteoarthritis using the Lakshmanan grading system in CBCT images. This study aims to address that gap by assessing incidental AOJ changes on CBCT and their relationship to age and sex. The severity of osteoarthritic changes was graded using the Lakshmanan classification system, and logistic regression models were employed to evaluate associations with demographic variables. Because these osteoarthritic changes were incidentally detected on CBCT scans performed for unrelated dental indications, this approach highlights the potential utility of CBCT in opportunistic diagnosis and early screening of clinically silent degenerative changes in the AOJ. In addition, the distribution of these changes according to age and sex was analysed. In this regard, the present study aimed to demonstrate the effectiveness of CBCT in detecting degenerative changes in the AOJ.

## MATERIALS AND METHODS

### Study design and ethical approval
This retrospective study was conducted in accordance with the principles of the Declaration of Helsinki and approved by the Karadeniz Technical University Faculty of Medicine's Ethics Committee (approval number: 2020/261, approval date: 12.10.2020). This study is a retrospective analysis of anonymized data. As no direct patient interaction occurred, the requirement for informed consent was not applicable.

### Patient selection
The imaging results of patients who were aged 18 years and older between 2018 and 2020 and who underwent CBCT were retrospectively reviewed. The indications for imaging included dental implants, orthodontic treatment planning, evaluation of impacted teeth and their relationships with neighbouring anatomical structures, temporomandibular joint problems, maxillofacial cysts and tumours, and the need to obtain a 3D model. CBCT images with evaluation-hindering artifacts were excluded.

Patients with a history of systemic diseases and conditions known to influence bone metabolism—such as diabetes mellitus, osteoporosis, and autoimmune disorders—were excluded to avoid confounding effects on cervical bone morphology. These conditions may independently alter the radiographic appearance of the AOJ and obscure the relationship between osteoarthritic changes and demographic factors such as age and sex.

To reduce potential selection bias, we retrospectively screened all 1,632 CBCT records available in the institutional imaging database during the study period. Of the 1,632 CBCT records initially screened, 1,417 were excluded for the following reasons: age under 18 years ($n = 82$), systemic or metabolic diseases affecting bone morphology ($n = 421$), artefactual CBCT images ($n = 310$), and incomplete visualization of the AOJ ($n = 604$). A flowchart

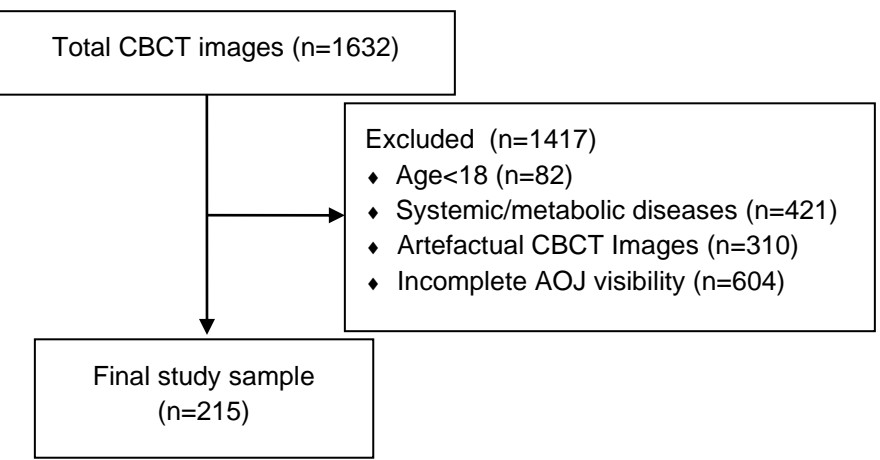

**Figure 1  Flowchart of the case selection process.** From an initial dataset of 1,632 CBCT images, 1,417 were excluded due to age under 18 ($n = 82$), presence of systemic or metabolic diseases ($n = 421$), image artifacts ($n = 310$), or incomplete visibility of the atlanto-odontoid joint ($n = 604$). The final study sample consisted of 215 eligible cases.

illustrating the application of the inclusion and exclusion criteria to the initial 1,632 records is presented in Fig. 1. Only those that met the eligibility criteria and provided complete visualization of the AOJ were included in the final sample.

Following the screening process, a total of 215 patients (130 females and 85 males) between the ages of 18 and 80 years who met the inclusion criteria and had CBCT images of the entire AOJ and associated bone structures. The patients were divided into six groups according to their age (18–29, 30–39, 40–49, 50–59, 60–69 and 70–80 years). These decade-based intervals were chosen in line with prior musculoskeletal and radiologic studies to facilitate age-stratified statistical comparisons and to ensure adequate sample size within each category.

## Imaging protocol

All of the CBCT images were obtained with a Kodak CS 9500 (Cone Beam 3D System; Kodak Dental Systems, Carestream Health, Inc., NY, USA). The scanning parameters were as follows: 85 kVp, 4.0 mA, 4 s scan time, voxel size ranging from 0.25 to 0.3 mm depending on the selected field of view, and 170 × 135–110 field of view. Patients were positioned with the Frankfort horizontal plane parallel to the floor and the midsagittal plane perpendicular to the floor, using adjustable head supports to ensure stability and minimize motion artifacts during scanning. The CBCT images were converted to DICOM format and transferred to the CS 3D Imaging Software (Carestream Dental LLC, Atlanta, GA, USA).

## Evaluation criteria

All CBCT images were reviewed using CS 3D Imaging Software (Carestream Dental LLC, Atlanta, GA, USA). Two experienced dentomaxillofacial radiologists (RBP and ÖSS) independently evaluated the axial and sagittal sections. The scans were presented in a
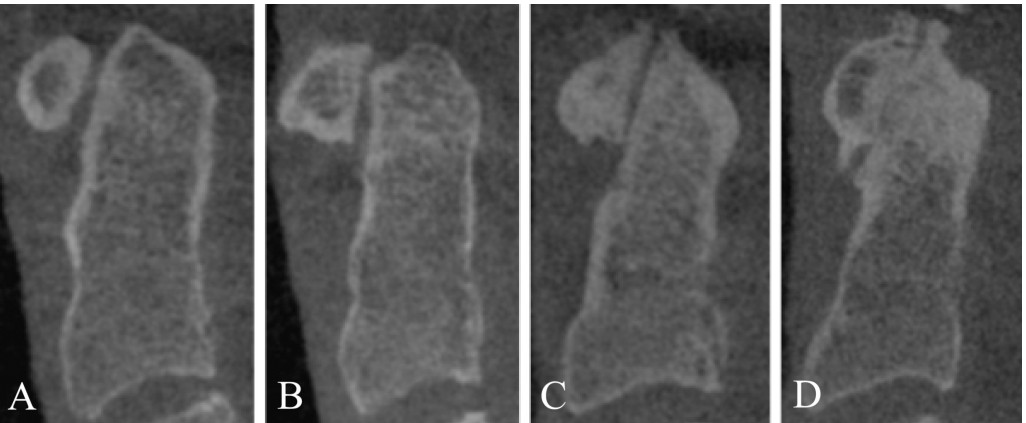

**Figure 2** **Severity of osteoarthritis. Sagittal view of the AOJ.** Grade 0: No osteoarthritis; normal AOJ (A). Grade 1: Mild osteoarthritis of the AOJ (B). Grade 2: Moderate osteoarthritis of the AOJ (C). Grade 3: Severe osteoarthritis of the AOJ (D).

randomized order, and both observers were blinded to each other's evaluations and to patient demographic data. Discrepancies were resolved by consensus. Prior to the formal evaluation, both radiologists jointly reviewed a pilot set of 15 CBCT scans using the Lakshmanan grading system to establish consistency in interpretation and standardize the application of the criteria. The inter-observer agreement was calculated based on the independent evaluations prior to consensus. To assess intraobserver reliability, RBP re-evaluated a randomly selected subset of 65 patients (approximately 30% of the total sample) one month after the initial evaluation.

The osteoarthritis in the AOJ was graded according to the classification described by *Lakshmanan et al. (2005)* (Figs. 2 and 3):

- Grade 0: Normal AOJ. A normal joint space with no osteophytes.
- Grade 1: Mild osteoarthritis. A narrowed AOJ space or a normal AOJ space with osteophytes. The osteophytes do not exceed either end of the atlas or apex of the dens.
- Grade 2: Moderate osteoarthritis. An obliterated joint space independent of the presence of osteophytes or the osteophytes exceed either end of the atlas or apex of the dens and have a height of <three mm.
- Grade 3: Severe osteoarthritis. Joint ankylosis with articular excrescences and/or transverse ligament calcification or the height of the osteophytes is >three mm.

Patients who were scored as Grade 0 were categorised as not having osteoarthritis, while those scored as Grades 1–3 were classified as having osteoarthritis.

## Statistical analysis

The data analysis was performed using IBM SPSS Statistics version 23 (IBM Corp., Armonk, NY, USA). The normality of the data was evaluated using the Shapiro–Wilk test. Ordinal logistic regression analysis was employed to investigate the factors affecting the severity of osteoarthritis. Binary logistic regression analysis was utilised to examine

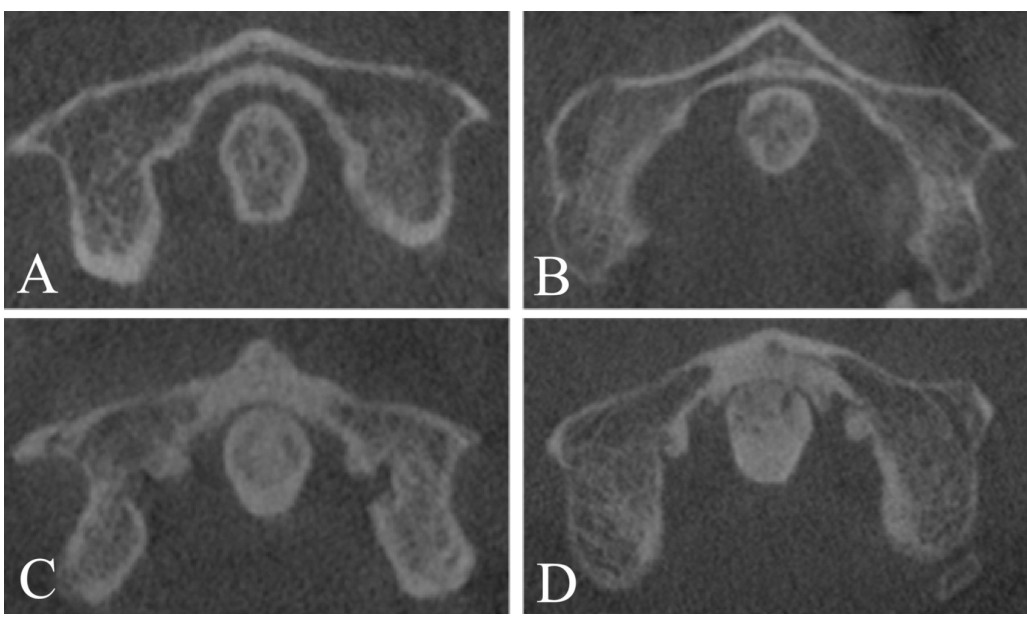

**Figure 3** **Severity of osteoarthritis. Axial view of the AOJ.** Grade 0: No osteoarthritis; normal AOJ (A). Grade 1: Mild osteoarthritis of the AOJ (B). Grade 2: Moderate osteoarthritis of the AOJ (C). Grade 3: Severe osteoarthritis of the AOJ (D).

the factors influencing the presence of osteoarthritis. Categorical data were compared using the Fisher–Freeman–Halton test, while multiple comparisons were analysed with the Bonferroni-corrected Z-test. Both intra- and inter-observer agreement was assessed using the Cohen's kappa coefficient. The results were considered statistically significant if $p < 0.05$.

## RESULTS

This study included 215 patients (130 females and 85 males) with a mean age of $38.97 \pm 15.82$ years (range: 18–79 years). The mean age of the female population was $39.15 \pm 15.48$ years (range: 19–78 years), while the mean age of the male population was $38.69 \pm 16.43$ years (range: 18–79 years). The prevalence and severity of osteoarthritis in the AOJ were analysed according to age and sex. Findings consistent with osteoarthritis in the AOJ were detected in 64.7% of patients ($n = 139$; 95% CI [58.2–71.2]), with a prevalence of 62.3% in females ($n = 81$; 95% CI [53.9–70.7]) and 68.3% in males ($n = 58$; 95% CI [58.3–78.3]).

The distribution of the osteoarthritis grades was as follows: Grade 0 in 35.3% of patients ($n = 76$), Grade 1 in 21.4% ($n = 46$), Grade 2 in 38.6% ($n = 83$) and Grade 3 in 4.7% ($n = 10$). Among the males, the most common grade was Grade 2 (47.1%, $n = 40$), whereas the females included a higher proportion of Grade 1 patients (25.4%, $n = 33$) (Fig. 4). Osteoarthritis severity increased significantly with age ($p < 0.001$), with higher grades predominantly observed in the older age groups, especially after the age of 50. In the 18–29 age group, 60.5% of patients ($n = 49$) were classified as Grade 0, while in the 70–80 age

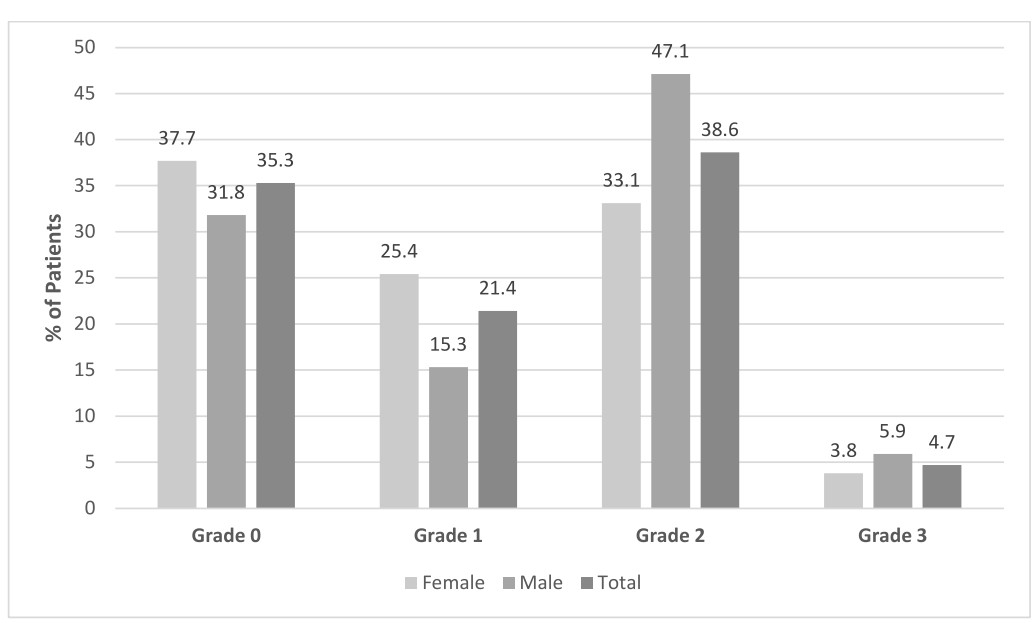

**Figure 4  Distribution of the scores for osteoarthritis in the AOJ by gender and overall.**

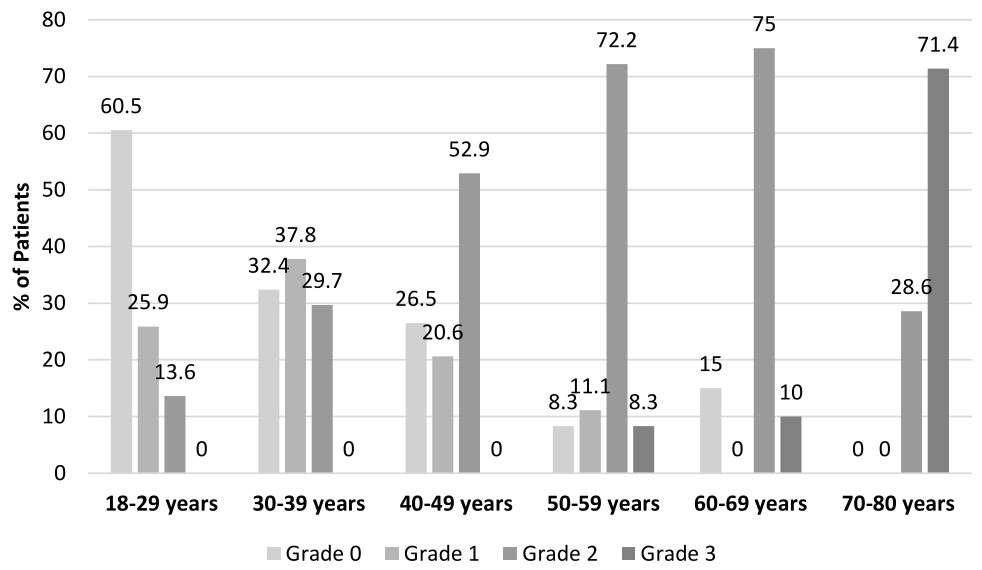

**Figure 5  Distribution of the scores for osteoarthritis in the AOJ by age group.**

group, 71.4% ($n = 5$) were classified as Grade 3 (Figs. 4 and 5). There was a statistically significant difference between the age groups and the osteoarthritis scores ($p < 0.001$). The distribution of the osteoarthritis scores by age group is shown in Table 1.

The ordinal logistic regression analysis revealed a statistically significant effect of sex on the severity of osteoarthritis. Females showed 44.3% decreased odds of severe osteoarthritis

**Table 1  Severity of osteoarthritis in the atlanto-odontoid joint across different age groups.**

| Age groups | Osteoarthritis scores n, (%) | | | | Test statistics | p |
|---|---|---|---|---|---|---|
| | Normal | Mild | Moderate | Severe | | |
| 18–29 | 49 (60.5)[a] | 21 (25.9)[a,b] | 11 (13.6)[a] | 0 (0)[a] | | |
| 30–39 | 12 (32.4)[a,b] | 14 (37.8)[b] | 11 (29.7)[a,b] | 0 (0)[a] | | |
| 40–49 | 9 (26.5)[b] | 7 (20.6)[a,b] | 18 (52.9)[b,c] | 0 (0)[a] | | |
| 50–59 | 3 (8.3)[b] | 4 (11.1)[a,b] | 26 (72.2)[c] | 3 (8.3)[a] | 103.983 | <0.001 |
| 60–69 | 3 (15)[b] | 0 (0)[a] | 15 (75)[c] | 2 (10)[a] | | |
| 70–80 | 0 (0)[b] | 0 (0)[a,b] | 2 (28.6)[a,b,c] | 5 (71.4)[b] | | |

Notes.
*Fisher-Freeman-Halton.
[a–c]There is no difference between groups with the same letters.
Bold values indicate statistically significant results at $p < 0.05$.

compared to males (odds ratio (OR) = 0.557, 95% confidence interval (CI) [0.318–0.975]; $p = 0.041$). While this sex-based difference in osteoarthritis severity reached statistical significance, its clinical implications could not be determined within the scope of this study, which did not include symptom-related or functional outcome data. Additionally, the age group showed a significant association with the severity of osteoarthritis. With the 70–80 age group serving as a reference, the odds of having less severe osteoarthritis were significantly higher in the younger age groups. Specifically, individuals aged 18–29 showed 99.9% decreased odds of severe osteoarthritis (OR = 0.001, 95% CI [0.0–0.008]; $p < 0.001$), those aged 30–39 showed 99.7% decreased odds (OR = 0.003, 95% CI [0.001–0.025]; $p < 0.001$), those aged 40–49 showed 99.4% decreased odds (OR = 0.006, 95% CI [0.001–0.045]; $p < 0.001$), those aged 50–59 showed 97.1% decreased odds (OR = 0.029, 95% CI [0.004–0.206]; $p < 0.001$), and those aged 60–69 showed 96.0% decreased odds of severe osteoarthritis compared to individuals aged 70–80 (OR = 0.040, 95% CI [0.005–0.308]; $p = 0.002$). The results of the ordinal logistic regression analysis are presented in Table 2.

The factors influencing the presence of osteoarthritis were analysed by means of binary logistic regression. The univariate analysis revealed no statistically significant effect of sex on the presence of osteoarthritis ($p = 0.375$); however, the risk of osteoarthritis being present increased by 7.9% for each additional year of age (OR: 1.079; 95% CI [1.052–1.107]; $p < 0.001$).

In the multiple logistic regression model, sex again showed no statistically significant effect on the presence of osteoarthritis ($p = 0.248$). Conversely, age remained a significant factor, with the presence of osteoarthritis increasing by 8% for each additional year of age (OR: 1.080; 95% CI [1.052–1.108]; $p < 0.001$). The results of the ordinal logistic regression analysis are shown in Table 3.

After the re-evaluation of the CBCT images, Cohen's Kappa coefficient was calculated as 0.884 (95% CI [0.813–0.955]), indicating excellent agreement between the two observations. In addition, intraobserver reliability was assessed by having RBP re-evaluate a randomly selected subset of 65 patients (approximately 30% of the sample) one month

**Table 2 Odds ratios for the association between sex, age groups, and increasing severity of osteoarthritis in the atlanto-odontoid joint.**

|  | OR (95% CI) | p |
|---|---|---|
| Sex |  |  |
| Female | 0.557 (0.318–0.975) | **0.041** |
| Male | Reference |  |
| Age groups |  |  |
| 18–29 | 0.001 (0–0.008) | **<0.001** |
| 30–39 | 0.003 (0–0.025) | **<0.001** |
| 40–49 | 0.006 (0.001–0.045) | **<0.001** |
| 50–59 | 0.029 (0.004–0.206) | **<0.001** |
| 60–69 | 0.040 (0.005–0.308) | **0.002** |
| 70–80 | Reference |  |

Notes.
Test of parallel lines ($\chi^2 = 73.712$; $p = 0.064$), OR (95% CI): Odds ratio (95% confidence interval).
Bold values indicate statistically significant results at $p < 0.05$.

**Table 3 Factors influencing the presence of osteoarthritis in the atlanto-odontoid joint: results from binary logistic regression analysis.**

|  | Univariate | | Multiple | |
|---|---|---|---|---|
|  | OR (95% CI) | p | OR (95% CI) | p |
| Sex |  |  |  |  |
| Female | 0.770 (0.432–1.372) | 0.375 | 0.681 (0.356–1.306) | 0.248 |
| Male | Reference |  |  |  |
| Age | 1.079 (1.052–1.107) | **<0.001** | 1.080 (1.052–1.108) | **<0.001** |

Notes.
Bold values indicate statistically significant results at $p < 0.05$.

later. This analysis yielded a Cohen's kappa coefficient of 0.908 (95% CI [0.781–1.000]), demonstrating excellent intraobserver agreement.

## DISCUSSION

This study revealed that osteoarthritic changes in the AOJ were prevalent in 64.7% of patients, with severity increasing with age and being more pronounced in males. Osteoarthritis in the AOJ was assessed using a CBCT-based grading system incorporated joint space narrowing and osteophyte formation in axial and sagittal views. Different imaging techniques have previously been used to study osteoarthritis in the AOJ (*Liu et al., 2014*; *Zapletal et al., 1997*; *Zapletal et al., 1995*; *Badve et al., 2010*). *Zapletal et al. (1995)* reported that the lateral cervical projection technique was particularly useful in imaging severe osteoarthritis in the AOJ, although CT provided the best radiographic detail for an accurate diagnosis. *Genez et al. (1990)* reported that osteophytes, cysts and transverse ligament calcifications on the median surface of the C1 vertebra were only detected by CT and that these structures were best observed in axial sections. *Zapletal et al. (1997)* also reported that T1-weighted MRI effectively demonstrated pathological changes in the AOJ. When compared with CT, CBCT offers a lower radiation dose and serves as

an effective imaging technique for evaluating small joint structures, including the AOJ (*Patcas et al., 2013*; *Takahata et al., 2018*). A recent study reported that the mean effective radiation dose for CBCT was 1.9 mSv, compared to 2.7 mSv for medical CT, representing an approximate 30% reduction in patient exposure, thus supporting CBCT's advantage in radiation safety (*Khader et al., 2024*). Although our study did not directly compare imaging modalities, recent findings have shown that CBCT provides diagnostic accuracy comparable to CT and MRI in detecting osseous changes, while offering advantages in accessibility and cost-effectiveness. These features support the practical use of CBCT in evaluating anatomical structures such as the AOJ in routine clinical settings (*Straub et al., 2024*). Degenerative changes in the AOJ were frequently observed as incidental findings on CBCT scans acquired for unrelated dental indications. The incidental detection of osteoarthritis in the AOJ during routine CBCT imaging highlights the importance of systematically evaluating the AOJ during dental radiographic examination. Given its wide availability, high spatial resolution, and lower radiation exposure, CBCT can be a practical tool in detecting early degenerative changes in the AOJ, especially when such findings are incidentally noted during dental evaluations.

The literature revealed that osteoarthritis is more commonly seen in females (*Alzahrani et al., 2020*; *Dodge, Mikkelsen & Duff, 1970*; *Kellgren & Lawrence, 1958*; *Prieto-Alhambra et al., 2014*; *Zhao et al., 2011*); however, there is conflicting data regarding the relationship between sex and osteoarthritis severity. For instance, *Srikanth et al. (2005)* reported that osteoarthritis of the knee was more severe in females, while there was no sex-based difference in the hand and hip joints. *Lawrence, Bremner & Bier (1966)* reported that the severity of osteoarthritis in the lumbar and cervical vertebrae did not vary according to sex, while *Wilder, Fahlman & Donnelly (2011)* reported that osteoarthritis in the cervical vertebrae progressed more rapidly in males than in females. Both *Liu et al. (2014)* and *Harata, Tohno & Kawagishi (1981)* reported that there was no sex-based difference in the prevalence of osteoarthritis in the AOJ. In our study, although the overall prevalence of AOJ osteoarthritis did not significantly differ by sex, males were more likely to exhibit severe forms of the disease. This finding may suggest that sex-related anatomical, hormonal, or biomechanical factors influence disease progression rather than its initial development. Recent biomechanical modeling has confirmed that sex-specific anatomical variations in cervical vertebrae geometry result in differential load distributions, with males generally bearing greater axial forces, potentially contributing to increased severity of AOJ degeneration (*Purushothaman & Yoganandan, 2022*).

Prior studies have shown that degenerative changes in the AOJ are seen in patients of older ages 60s' and 70s' (*Liu et al., 2014*; *Harata, Tohno & Kawagishi, 1981*). *Betsch et al. (2015)* revealed that the prevalence of osteoarthritis in the AOJ increased significantly with age, ranging from 1.4% in the youngest age group to 92.9% in the oldest age group. *Von Torklus & Gehle (1972)* detected osteoarthritis in the AOJ in 36% of patients aged 41–50 years and 88% of patients over 60 years of age. This study demonstrated that osteoarthritis in the AOJ is more prevalent and more severe in males, which contrasts with findings concerning other joints, such as the knee, where females typically exhibit greater severity (*Alzahrani et al., 2020*; *Dodge, Mikkelsen & Duff, 1970*; *Kellgren & Lawrence, 1958*;

*Prieto-Alhambra et al., 2014*; *Zhao et al., 2011*). The increased severity seen in males may be due to differences in biomechanics or occupational exposure, which should be explored further in future research. Additionally, the progressive increase in osteoarthritis severity seen with age aligns with the findings of previous studies, underscoring the need for early diagnosis and management to prevent complications such as odontoid fractures.

Given its high prevalence among older adults, AOJ degeneration represents an increasingly relevant public health concern. Prior studies have shown that AOJ osteoarthritis may affect up to 93% of individuals over 70 years of age (*Liu et al., 2014*). Degenerative changes in the upper cervical spine can lead to reduced mobility, cervicogenic pain, and increased fall risk, which are all factors negatively impacting health-related quality of life (*Shalhoub et al., 2022*). Furthermore, the global burden of osteoarthritis—including cervical spine involvement—has been estimated to account for 1–2.5% of the gross national product in developed nations, driven by both direct healthcare costs and productivity loss (*Leifer, Katz & Losina, 2022*). These findings highlight the broader implications of AOJ degeneration and support the clinical value of early detection through accessible modalities such as CBCT.

A significant limitation of this study is its retrospective nature and its reliance on CBCT images taken for routine dental purposes, which limits the generalisability of the findings. Since the scans were not obtained from a general population-based sample but rather from individuals referred for specific dental imaging needs, the possibility of selection bias must be acknowledged. Additionally, although all eligible scans were screened consecutively, potential selection bias inherent to retrospective designs cannot be fully excluded. Furthermore, the absence of clinical symptom-related data represents a major limitation, as it prevents direct correlations from being drawn between radiographic findings and functional or pain-related outcomes. Moreover, as all patients were from a single country (Turkey), the demographic and ethnic generalizability of the findings to broader populations may be limited. Although *a priori* power analysis could not be conducted due to the retrospective design, the sample size ($n = 215$) and the large effect sizes observed in key comparisons (*e.g.*, age-related odds ratios ranging from 0.001 to 0.040) suggest that the study was sufficiently powered to detect meaningful associations. In addition, all image grading was conducted by radiologists trained in dentomaxillofacial radiology. While interobserver agreement was strong, this may limit the generalizability of the grading approach to other clinical disciplines, such as orthopedic radiology or rheumatology.

Future studies should include symptomatic patients and evaluate the clinical relevance of radiographic findings. Prospective studies with larger and more diverse populations could provide further insights into the progression of osteoarthritis in the AOJ and its clinical implications.

## CONCLUSION

Only a few studies found in the literature have investigated osteoarthritis in the AOJ. To the best of our knowledge, this is the first comprehensive study to evaluate osteoarthritis in

the AOJ by means of CBCT imaging. The results of this study highlight CBCT's potential as a valuable imaging modality due to its lower radiation doses when compared with conventional CT and its ability to effectively identify degenerative changes in the AOJ.

Given the retrospective nature of this study, the potential for selection bias, and the absence of clinical symptom data, the findings should be interpreted with caution. While the incidental detection of AOJ degeneration on CBCT may support early recognition, further validation in clinical settings is required before routine application can be recommended.

Further research is needed to investigate the correlation between radiographic findings and clinical symptoms in patients with osteoarthritis in the AOJ, as well as to validate the present findings in larger and more diverse populations. Additionally, prospective studies focusing on the progression of osteoarthritis in the AOJ and the development of improved severity scoring methods could provide deeper insights into the condition. Future studies may also explore the potential of artificial intelligence-based algorithms for the automated detection and grading of AOJ osteoarthritis on CBCT scans, which could enhance diagnostic efficiency and consistency in routine clinical practice.

**Abbreviations**

| | |
|---|---|
| **AOJ** | Atlanto-odontoid joint |
| **CBCT** | Cone-beam computed tomography |
| **C1** | First cervical vertebrae |
| **C2** | Second cervical vertebrae |
| **3D** | Three-dimensional |
| **MRI** | Magnetic resonance imaging |
| **OR** | Odds ratio |
| **CI** | Confidence interval |

## ACKNOWLEDGEMENTS

This study is based on the specialty thesis submitted by Ramazan Berkay Peker as part of the Dental Specialty Education Program in Dentomaxillofacial Radiology at Karadeniz Technical University, under the supervision of Assoc. Prof. Dr. Ömer Said Sezgin.

### Funding

The authors received no funding for this work.

### Competing Interests

The authors declare there are no competing interests.

### Author Contributions

- Ramazan Berkay Peker conceived and designed the experiments, performed the experiments, analyzed the data, prepared figures and/or tables, authored or reviewed drafts of the article, and approved the final draft.

- Omer Said Sezgin conceived and designed the experiments, performed the experiments, analyzed the data, prepared figures and/or tables, authored or reviewed drafts of the article, and approved the final draft.
- Saadettin Kayipmaz conceived and designed the experiments, prepared figures and/or tables, and approved the final draft.
- Senem Tuğra Dönmez conceived and designed the experiments, authored or reviewed drafts of the article, and approved the final draft.

### Human Ethics

The following information was supplied relating to ethical approvals (i.e., approving body and any reference numbers):

Ethical approval for this study was granted by the Karadeniz Technical University Faculty of Medicine Ethics Committee (Approval Number: 2020/261, Approval Date: 12.10.2020) in accordance with the principles of the Declaration of Helsinki.

### Data Availability

The raw measurements are available in the Supplementary Files.

### Supplemental Information

Supplemental information for this article can be found online at http://dx.doi.org/10.7717/peerj.19569#supplemental-information.

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
