# Peer review of "Assessment of degenerative changes in the atlanto-odontoid joint using cone-beam computed tomography (CBCT) imaging"

_PeerJ, doi:10.7717/peerj.19569_

## Round 0.1 · original submission · Major Revisions

Please take care to address the comments from Reviewers 1 and 3.

Reviewer 1 ·

Basic reporting

1. General Assessment
This study investigates the severity of osteoarthritis (OA) in the atlanto-odontoid joint (AOJ) and its relationship with age and sex using cone-beam computed tomography (CBCT). The study highlights CBCT’s potential for detecting degenerative changes in the AOJ, which could contribute to clinical decision-making. However, several methodological, statistical, and interpretative aspects require revision to enhance the study’s clarity, robustness, and scientific contribution.

Experimental design

2. Areas for Improvement and Recommendations
A) Language and Structural Clarity
While the manuscript is generally well-written, some sections require refinements to improve readability and precision.

Phrases such as "Females exhibited a 44.3% lower likelihood of severe osteoarthritis in the AOJ when compared with males" could be reworded for better fluency.
Statistical findings should be presented more clearly to facilitate reader comprehension.
Recommendation: A thorough English language review by a native speaker or professional editing service is advised.

B) Literature Review and Introduction
The introduction provides sufficient anatomical background and highlights the clinical importance of AOJ osteoarthritis. However, the discussion of CBCT’s advantages and disadvantages compared to other imaging modalities is underdeveloped.

More emphasis should be placed on CBCT’s role compared to conventional CT and MRI for AOJ assessment.
The literature review should integrate more recent studies, particularly those focusing on CBCT’s application in diagnosing AOJ osteoarthritis. Lakshmanan et al. (2005) and Betsch et al. (2015) are useful references, but newer studies should be included.
Recommendation: Expand the literature review to highlight CBCT’s specific advantages in AOJ imaging and incorporate studies from 2021 onwards to ensure relevance.

C) Methodology
The methods section is generally well-structured but lacks critical details:

Patient Selection Criteria:

While inclusion and exclusion criteria are stated, the rationale for excluding specific comorbidities (e.g., diabetes, osteoporosis) should be explained.
The selection of CBCT images appears random, but potential selection biases should be addressed.
Imaging Protocol:

The standardization of patient positioning during CBCT scans should be explicitly described.
Clarify how the two independent radiologists assessed the images, including the software used and the order of evaluation.
Observer Reliability:

Cohen’s kappa coefficient (0.884) indicates excellent interobserver agreement, but intraobserver reliability is not reported.
Recommendation: Provide additional details on image standardization, randomization procedures, and intraobserver reliability to strengthen the methodological rigor.

Validity of the findings

D) Results
The results section is well-structured and supported by tables. However, statistical findings should be interpreted more clearly.
The reported 44.3% lower likelihood of severe AOJ osteoarthritis in females (p = 0.041) requires further clarification:
Is this clinically significant, or merely statistically significant?
Why does overall OA prevalence not significantly differ by sex, while severe OA does?
Recommendation: The clinical implications of statistical findings should be better explained, particularly in terms of sex differences and their biomechanical or hormonal basis.
E) Discussion and Conclusion
The discussion effectively summarizes the study’s findings, but several aspects require further elaboration:

CBCT vs. Conventional Imaging: A more detailed comparison with CT and MRI should be included.
Limitations of the Study:
The retrospective design and potential selection biases should be explicitly acknowledged.
The lack of clinical symptom correlation with radiographic findings is a major limitation that should be discussed.
Sex-Based Differences: The higher severity of osteoarthritis in males should be explored further, considering biomechanical, hormonal, and occupational factors.
Recommendation: Enhance the discussion by addressing CBCT’s role in clinical decision-making, the study’s limitations, and possible mechanisms behind sex differences.

F) Reference Quality and Citations
A major concern is the lack of recent references:

The most recent citation is from 2020 (Alzahrani et al.), indicating that studies published in the last 3-5 years have not been incorporated.
Recent advances in CBCT analysis are missing.
Key issues:

Overreliance on older CT and MRI studies (e.g., Zapletal et al., 1995; Genez et al., 1990).
Limited references on CBCT’s diagnostic accuracy and radiation dose benefits.
No citations from 2021-2025, despite significant advancements in CBCT technology.
Recommendation: Conduct a systematic literature review to include the latest findings on CBCT applications in AOJ osteoarthritis, particularly those from 2021 onward.

Additional comments

4. Final Recommendation
This study has the potential to contribute valuable insights into the use of CBCT for AOJ osteoarthritis assessment. However, several major revisions are necessary before publication:

Expand the literature review to include recent studies (2021-2025).
Clarify methodology, particularly in patient selection, image standardization, and observer reliability.
Enhance the statistical interpretation and explain the clinical significance of findings.
Improve the discussion by providing a more detailed comparison of CBCT with conventional imaging and acknowledging study limitations.
Revise language and structure to improve clarity and readability.
Recommendation: Major Revision

·

Basic reporting

The article is well written with good English and well-structured, and with good acceptable standards.

Experimental design

The research has highlighted an uncovered area with original research. Well-defined research questions and a rigorous investigation were performed. The methodology is well executed.

Validity of the findings

The literature review is well done and has the highest novelty. the conclusion is well written.

·

Basic reporting

The manuscript investigates both the frequency and level of severity of atlanto-odontoid joint arthritis by conducting cone-beam computed tomography scans on dental patient records from the past. While the study addresses a clinically relevant and underexplored topic, the manuscript in its current form requires substantial revisions across all sections. The manuscript demands extensive improvements for both scientific and clinical purposes to boost its value.
Introduction
• The introduction should begin by elaborating on the clinical significance of cervical spine biomechanics and then transition to AOJ anatomy.
• Add biomechanical significance; clearly state how AOJ's 40-70% relates to cervical rotation and is prone to wear/tear. In addition, include current anatomical or biomechanical studies that confirm this wide range of rotation values (40–70%).
• Support public health claim: The statement about osteoarthritis in the AOJ being a "significant public health concern" would be stronger with incidence/prevalence data or citing epidemiological reviews.
• The statement of ‘public health concerns’ regarding osteoarthritis in the AOJ would be strengthened by providing incidence or prevalence rates or references to epidemiological studies.
• Strengthen symptom linkage: A stronger relationship between osteophytes and headaches through restricted movement should be defined through an analysis of neurovascular anatomy.
• Clarify advantages of CBCT: Mention its benefits, such as the CBCT technology requiring a reduced radiation dose compared to traditional methods, convenient availability in dental offices, or cost-efficient operation, to justify its investigation.
• Refine the research gap sentence by strengthening the phrasing to explicitly state what hasn't been studied, e.g., “The research lacks any prior investigation that measured AOJ osteoarthritis seriousness with the Lakshmanan grading system when using CBCT.”
• The idea of "incidental detection" is interesting and novel. Further emphasize and elaborate on its value for early screening or opportunistic diagnosis.
• The research objective should be more clearly detailed about the grading systems that were utilized, as well as highlight the implementation of logistic regression for analysis.

Experimental design

Materials and Methods
• Consider including a CONSORT-style flowchart to illustrate how 1,632 patients became 215 after the inclusion and exclusion criteria were applied.
• The researcher needs to clarify when the examination of inter-observer agreement took place, before or after the evaluation, and how radiologists managed their knowledge about patient demographics to minimize bias.
• The paper should provide reasoning about why specific age periods (decades) were selected as grouping criteria based on current research standards.
• Lakshmanan grading: Reproduce the full grading criteria (osteophyte size thresholds, joint space metrics) in-text rather than citing, as readers may lack access.
• A description regarding the process for radiologist training (for example, a pilot study involving 20 cases) to create a consistent grading system usage.
• The prevalence data (such as 64.7%) needs to include 95% confidence intervals to demonstrate precision in measurements.
• The term “progressively” needs clarification to verify whether the change occurs in a linear pattern or shows significant growth between selected age ranges.
• The author should simplify odds ratios by providing concrete explanations that connect statistical data to practical scenarios, such as "Females showed 44% decreased odds of severe OA compared to males."
• The inclusion of kappa confidence intervals enhances the interpretive value of the results.

Validity of the findings

Results:
• It is important to justify the reduction from 1,632 to 215 patients by specifying how many were excluded due to medical conditions and how many due to poor-quality or incomplete CBCT images.
• The authors need to address selection bias limitations in their research because the CBCT scans were initially obtained for dental purposes, thus affecting broader population generalization.
• Although a priori power analysis may not be feasible for retrospective studies, the inclusion of a post hoc power calculation or effect size estimation would enhance the statistical robustness and credibility of the findings.
• The manuscript must provide information about CBCT scan radiation dosages expressed in mSv to substantiate how it exposes patients to less radiation than conventional CT.
• The manuscript should clarify if investigators used a single or dual voxel dimension of “0.3–0.25 mm” in the study or if this feature changed according to patient conditions or imaging protocols.

Additional comments

Discussion
• The initial part of the discussion section must provide a summary of key research outcomes. The section about AOJ assessment within the second paragraph stands as the only needed content of this paragraph. Concisely summarize the second paragraph into one sentence to cover the main point about AOJ assessment and include it in the initial part of the discussion.
• The first paragraph of the discussion should briefly summarize the main results. The second paragraph in the discussion seems unnecessary except for expanding and rewriting the assessment of AOJ. Please condense the entire second paragraph into a single sentence and integrate it with the first paragraph.
• Consider elaborating on the finding that males experience a higher severity of AOJ osteoarthritis.
• AOJ degeneration is characterized as a public health concern in the introduction, yet the discussion fails to return to this question. The discussion should add epidemiological data and analyze both economic consequences and quality of life changes in older adult populations.
• No mention of limitations related to grading bias: Although interobserver agreement was strong, all grading was performed by radiologists trained in dentomaxillofacial imaging. This may limit generalizability to broader clinical settings—e.g., orthopedic radiology or rheumatology.
• The study needs to mention two more limitations, such as a lack of clinical symptom data and demographic homogeneity (all patients from Turkey).
• could be stronger, suggesting that automated detection using AI on CBCT could position the paper at the frontier of imaging research.
• Future directions could be stronger if they included artificial intelligence for automated detection on cone-beam computed tomography imaging.
Conclusion
The current conclusion may be interpreted as overstating the utility of CBCT for routine AOJ assessment. Given the incidental nature of detection and lack of clinical correlation, this recommendation should be tempered or framed as exploratory. Please include three primary limitations in your analysis: retrospective design, selection bias, and missing clinical data. Future studies need to unite automated diagnoses and clinical measurements and showcase the groundbreaking aspects within the presented data.

---

## Round 0.2 · accepted · Accept

The manuscript can be accepted in its current form.

Reviewer 1 ·

Basic reporting

I would like to commend you on the improvements made to the article in response to the deficiencies noted in the initial review. The revisions have addressed the concerns effectively, and the manuscript is now in excellent form.

After careful consideration, I believe that the article is now suitable for publication. I would like to thank you for your time and effort in enhancing the manuscript, and I am confident that it will contribute significantly to the field.

Wishing you all the best for the publication process.

Kind regards,

Experimental design

.

Validity of the findings

.

Additional comments

.

·

Basic reporting

N/A

Experimental design

N/A

Validity of the findings

N/A

Additional comments

Dear Authors,

The authors have addressed all the comments and suggestions, and the manuscript has dramatically improved. I want to congratulate the authors and wish them all the very best in their future endeavours.

Best regards and keep well